# Toward the Super Temporal Resolution Image Sensor with a Germanium Photodiode for Visible Light

**DOI:** 10.3390/s20236895

**Published:** 2020-12-02

**Authors:** Nguyen Hoai Ngo, Anh Quang Nguyen, Fabian M. Bufler, Yoshinari Kamakura, Hideki Mutoh, Takayoshi Shimura, Takuji Hosoi, Heiji Watanabe, Philippe Matagne, Kazuhiro Shimonomura, Kohsei Takehara, Edoardo Charbon, Takeharu Goji Etoh

**Affiliations:** 1School of Science and Engineering, Ritsumeikan University, 1-1-1 Noji-Higashi, Kusatsu, Shiga 525-8577, Japan; gr0397xr@ed.ritsumei.ac.jp (N.H.N.); skazu@fc.ritsumei.ac.jp (K.S.); 2School of Electronics and Telecommunications, Hanoi University of Science and Technology, 1 Dai Co Viet, Bach Khoa, Hai Ba Trung, Hanoi 100803, Vietnam; quang.nguyenanh@hust.edu.vn; 3IMEC, 3001 Leuven, Belgium; Fabian.Bufler@imec.be (F.M.B.); Philippe.Matagne@imec.be (P.M.); 4Faculty of Information Science and Technology, Osaka Institute of Technology, Hirakata Campus, 1-79-1 Kitayama, Hirakata City, Osaka 573-0196, Japan; yoshinari.kamakura@oit.ac.jp; 5Link Research Corporation, 291-4, Kuno, Odawara-shi, Kanagawa 250-0055, Japan; hideki.mutoh@nifty.com; 6Graduate School of Engineering, Osaka University, 1-1 Yamada-oka, Suita, Osaka 565-0871, Japan; shimura@prec.eng.osaka-u.ac.jp (T.S.); hosoi@prec.eng.osaka-u.ac.jp (T.H.); watanabe@prec.eng.osaka-u.ac.jp (H.W.); 7Department of Civil and Environmental Engineering, School of Science and Engineering, Kindai University, 3-4-1 Kowakae, Higashi-Osaka, Osaka 577-8502, Japan; takehara@civileng.kindai.ac.jp; 8Advanced Quantum Architecture Laboratory, EPFL, Rue de la Maladière 71b, Case Postale 526, CH-2002 Neuchâtel, Switzerland; edoardo.charbon@epfl.ch

**Keywords:** ultra-high-speed, super temporal resolution, temporal resolution limit, image sensor, visible light, SWIR, germanium

## Abstract

The theoretical temporal resolution limit $t_{T}$ of a silicon photodiode (Si PD) is 11.1 ps. We call “super temporal resolution” the temporal resolution that is shorter than that limit. To achieve this resolution, Germanium is selected as a candidate material for the photodiode (Ge PD) for visible light since the absorption coefficient of Ge for the wavelength is several tens of times higher than that of Si, allowing a very thin PD. On the other hand, the saturation drift velocity of electrons in Ge is about 2/3 of that in Si. The ratio suggests an ultra-short propagation time of electrons in the Ge PD. However, the diffusion coefficient of electrons in Ge is four times higher than that of Si. Therefore, Monte Carlo simulations were applied to analyze the temporal resolution of the Ge PD. The estimated theoretical temporal resolution limit is 0.26 ps, while the practical limit is 1.41 ps. To achieve a super temporal resolution better than 11.1 ps, the driver circuit must operate at least 100 GHz. It is thus proposed to develop, at first, a short-wavelength infrared (SWIR) ultra-high-speed image sensor with a thicker and wider Ge PD, and then gradually decrease the size along with the progress of the driver circuits.

## 1. Introduction

The theoretical temporal resolution limit of a silicon photodiode (Si PD) is 11.1 ps (90.1 Gfps) [1]. However, users in advanced sciences require much faster multiframing than this limit. For example, a pump-probe method is a common technique to measure phenomena changing in several femtoseconds to nanoseconds. If the change is captured with one-shot imaging, the measurement performance is significantly improved. Furthermore, the pump-probe method can be applied to only reversible phenomena. This paper is a pilot study toward such an ultrafast continuous imaging. We propose the definition of “super temporal resolution (STR)” as a temporal resolution less than the theoretical limit of the Si PD since silicon image sensors are commonly used for high-speed imaging.

The triangles in Figure 1 show imaging devices having achieved STR [2,3,4,5,6,7,8,9,10,11,12,13,14]. Some of which are presented with motion pictures on the page for Invited Speakers on the webpage of ICHSIP31 [15].

In 1991, the frame rate of the fastest image sensor in the world was 4500 fps [16], have exponentially increased [17,18,19,20,21], and, in 2018, it reached 100 Mfps (equivalent to the frame interval of 10 ns) [22,23,24,25,26,27]. For X-ray imaging, a temporal resolution of 3 ns (333 Mfps) has been achieved using a silicon image sensor [23,27]. The large number of electrons generated by a single X-ray photon allows an increase of the frame rate without the suppression of the noise associated with the conversion of electron packets to voltage signals at an extremely high operating rate. For visible light, the current highest frame rate is about 100 Mfps.

Numerous devices have been developed for high-speed imaging. However, once an image sensor has achieved a high frame rate, other devices developed for imaging for a similar frame rate were immediately replaced by a camera with the image sensor. Except silicon image sensors, there still remain the imaging devices which achieved the STR. The frame rate of about 50 ps may be achieved by an image sensor with existing technologies [25]. This is close to the theoretical limit. Now, it is worth exploring image sensor technologies to go beyond the theoretical limit of the Si PD.

Germanium is a promising material for a high-speed transistor and for a photodiode in an SWIR range [28,29,30,31,32,33,34]. Thermal imaging together with motion imaging is very important in ultra-high-speed imaging. For example, in the automobile industry, high-speed imaging of heat generation provides useful information for improving the combustion efficiency of engines, tire wheels at sudden braking, a blast of an airbag, and laser and plasma processes in machining. However, the highest frame rate of existing IR video cameras is only 10 kfps for a reasonable pixel count. The nominal highest frame rates in the catalogues are about 100 kfps. However, the pixel count for the frame rate is impractically small.

STR imaging with a Ge PD for visible light arose in our discussion on the development of a high-speed SWIR image sensor. For a Si PD, the temporal resolution limit is approximately proportional to the ratio of the penetration depth δ of light to the saturation drift velocity $v_{c}$ [1]. The penetration depth of visible light to Ge is 1/50 to 1/90 of that to Si with a large variation depending on the source literature, and the saturation drift velocity $v_{c}$ in Ge is about 2/3 of that in Si. Therefore, it is expected that the temporal resolution limit of the Ge PD for the visible light is less than 1/30 (≈1/50 × 3/2) of the theoretical limit of the Si PD.

This article is organized as follows. Definitions on the temporal resolution are summarized. Tools for the analyses are explained, including the exact formulation based on the drift-diffusion equation and the flux equation (the DDF model), the approximate explicit expression derived by us [1] with their asymptotic expressions for the infinitesimal thickness of the photodiodes, and the Monte Carlo simulation codes. Models and the parameters for the analyses are summarized. Then, the temporal resolutions of the Si PD and the Ge PD for visible light are analyzed. The Si PD has been analyzed by our inhouse simulation code for Si semiconductors. For the analysis of the Ge PD, a Sentaurus MC simulation code by Synopsys is introduced. However, standard deviations of the propagation times of charges between electrode pairs are not included in the output list, which are the basis to evaluate the temporal resolution. Therefore, a method to estimate the temporal resolution with the Sentaurus code was proposed. To keep consistency of the analyses, both simulation codes are applied to a very thin Si PD to confirm agreement of the results. Then, the temporal resolution of the Ge PD for visible light are evaluated with the proposed method.

## 2. Definitions of Temporal Resolution and Tools for Analyses

### 2.1. Criteria for Temporal Resolution

The spread of the arrival time of signal electrons to a detection plane is represented by the standard deviation σ. As shown in Figure 2, the sum of the ordinates of two Gaussian distributions has no separable peak for $\mu_{2\;}-\mu_{1}<2\sigma$, where $\left(\mu_{2}>\mu_{1}\right)$, and $\mu_{2}\;\mathrm{and}\;\;\mu_{1}$ are the averages of the Gaussian distributions. Physically, the time difference of a pair of incident light pulses to the backside of an image sensor is kept as the difference of the averages of the arrival times as shown in Figure 3. Therefore, we defined the theoretical temporal resolution limit as $t_{T}=2\sigma$. However, 16% of the distribution function overlaps the next one. Therefore, we also propose another criterion to define the practical temporal resolution $t_{P}=t_{95}-t_{5}$ where $t_{95}\;\mathrm{and}\;t_{5}$ are, respectively, the times when 95% and 5% of the signal electrons arrive at the detection plane. For the Gaussian distribution, $t_{P}=3.30\sigma$. Empirically, this relationship fits well even for a skewed distribution of the arrival time with a long tail.

To evaluate the temporal resolution, the thickness of the photodiode needs to be specified. For the theoretical analysis, we employed a thickness $W_{T}=\delta$, where δ is the average penetration depth, 1/(absorption rate α), of the incident light. For this condition, 63.2% of incident photons are converted to electron-hole pairs, and 36.8% are passing through the photodiode. Ultra-high-speed imaging requires the highest possible sensitivity. Therefore, we also use another thickness, $W_{P}=3\delta$, for practical evaluation for which 95.0% of the incident photons undergo photoelectron conversion.

Therefore, in this article, the temporal resolution is evaluated for a combination of the thickness of the photodiode and the standard deviation of the arrival time; $\left(W_{T},\;t_{T}\right)$ = $\left(\delta ,\;2\sigma \right)$ and $\left(W_{P},\;t_{P}\right)$ = $\left(3\delta ,\;3.3\sigma \right)$ for the theoretical and the practical evaluations.

### 2.2. Temporal Resolution Based on DDF Models

The standard deviation of the arrival time can be exactly formulated based on the drift-diffusion equation and the flux equation. Hereafter, the model expressed by the set of the equations is called the DDF model. For a simple model as shown in Figure 4, the standard deviation is expressed with a set of equations shown in Appendix A. The expression requires numerical integration to obtain the standard deviation.

Etoh et al. derived the approximate explicit expression which accurately agrees with the numerical result of the DDF model except for a very thin photodiode [1]. The approximate expression is shown in Appendix B.

The thickness of the Ge PD for visible light is extremely thin due to much higher absorption coefficient of Ge than that of Si. Therefore, we also derived the asymptotic expressions for the infinitesimal thickness *W* of the photodiode. For the approximate expression, the asymptotic expressions can be derived as shown in Appendix C. Analytical derivation of the limit may be possible for the infinitesimal W. For the DDF model, we found the asymptotic expression through numerical integration of the exact formulation for various parameter values. The standard deviation always converges to $\sqrt{5}D_{c}/v_{c}^{2}$, which is physically incorrect. It may be due to the expression formulated with continuous variables, while the stepwise motion of the electrons becomes significant in such a thin layer.

### 2.3. Monte Carlo Simulation Codes

The Monte Carlo (MC) simulation code for Si used in this paper is an inhouse one developed by Kunikiyo, Kamakura et al. in Osaka University for homogeneous intrinsic silicon [35]. The model employs realistic physical models for (1) the energy band structure, (2) the phonon dispersion relations, (3) the electron-phonon scattering rate, (4) the impact ionization rate, and (5) the conservation of both energy and momentum in each scattering process to properly represent the final state of the scattering electron. The model requires no adjustable parameters and the results quantitively agree to the experimental data.

We do not have an MC simulation code for Ge. Therefore, we employed the Sentaurus MC simulation code marketed by Synopsys. The code is applicable to a Si_1−x_Ge_x_ compound, where x denotes the ratio of Ge. However, it is not directly applicable for our analysis. For example, the standard deviation to estimate the temporal resolution is not included in the output list, and the source of electrons cannot be placed in the bulk. The program can be easily modified, but the source program is not open. Therefore, we developed a method to utilize the code for our purpose without modification of the code (Appendix D).

## 3. Numerical Analyses

### 3.1. Model Parameters

Figure 5 shows the dependence of the drift velocity and the longitudinal diffusion coefficient on the electric field for Si and Ge [36,37,38]. The temperature is mostly 300 K. We defined the critical field as the field at which the drift velocity is 95% of the saturation value, since the saturation value keeps elongated at a constant value for a wide range of fields above their critical values (truncated in Figure 5). Table 1 shows for Si and Ge, (1) critical field, (2) drift velocity and diffusion coefficient for the critical field, and (3) penetration depth of light of wavelengths of 550 nm and 1 μm [39,40].

Average penetration depths (absorption coefficients) largely vary, depending on the source in the literature. For example, the penetration depth of light at 550 nm in Si can be 1.08, 1.44, and 1.73 μm, depending on the references [39,40,41]. We employed the value 1.73 μm from the latest reference [39], providing the largest standard deviation and the safety-side estimation of the temporal resolution limit.

For Ge, some sources are available for the absorption coefficients in the SWIR range. However, we found only one data set for visible light in [40] (pp. 465–478) published in 1985. The absorption coefficient based on the data set is easily extracted by the code supplied by [42]. Figure 6 seems to be generated from the same source and copied several times by many people. If we calculate the penetration depth of Ge for 550 nm directly from [40] or [42], the value is 20.0 nm, which is employed in this paper. However, from Figure 6, it is 26.7 nm, which may be due to rounding during multiple copying.

Considering the distributions of data on the penetration depths in the literature, the ratio of the penetration depth of Ge to Si varies from about 1/50 to 1/90. Therefore, the values presented in the conclusions of this paper are the tentative values. Accurate data for the penetration depths are expected to be specified by a professional society.

From Table 1, the ratio of the penetration depth for Ge and Si for light at 550 nm is 1/86.5 (=0.020/1.73). The critical field of Ge, 4.25 kV/cm, is about 17% of that of Si, 25 kV/cm. The saturation velocities and the dielectric constants of Ge are, respectively, about 2/3 and 3/2 of those of Si. The diffusion coefficient for Ge is about four times larger than that of Si.

### 3.2. Analysis for Si PD

In Figure 7, the green line $\sigma_{ddf}$ shows the standard deviation of the arrival time calculated by numerical integration of the expression for the DDF model shown in Appendix A with the parameters shown in Table 1. The orange dotted line $\sigma_{app}$ shows the approximate expression of the standard deviation shown in Appendix B with the mixing component $\sigma_{mix}$, that is due to the distribution of the penetration depth of incident light and the drift velocity, and the diffusion component $\sigma_{diff}$. The figure suggests that:

there are three ranges: the mixing (drift) effect is dominant for 0.14 μm<W<100 μm, which covers the whole practically meaningful range, and the diffusion effect is dominant for the very thin range W<0.14 μm and for the very thick range W>120 μm (not drawn);for W>0.14 μm, the approximate expressions almost perfectly agree with the exact solution from the DDF model; for W<0.14 μm, the discrepancy increases and the DDF model asymptotically converges to a constant value, $\sigma_{ddfL}=\sqrt{5}D_{c}/v_{c}^{2}$;the range used in practical applications, δ<W<3 δ, is included in the drift-dominant range as concluded in our previous paper [1];The theoretical temporal resolution limit for $\left(W_{T},\;t_{T}\right)$ is $\left(\delta ,\;2\sigma \right)=\left(1.73\;\mu m,\;11.1\;\mathrm{ps}\right)$, and the practical limit for $\left(W_{P},\;t_{P}\right)$ is $\left(3\delta ,\;3.3\sigma \right)=\;\left(5.19\;\mu m,\;45.2\;\mathrm{ps}\right)$.

Since the penetration depth of visible light is more than 1 μm, except for light close to the UV region, our approximate expression can be used for estimation of the temporal resolution limit for the thickness range. However, the penetration depth of 550 nm light to Ge is 20.0 nm. For W < 1 μm, these values significantly deviate from each other. Therefore, we evaluated the standard deviation of the arrival time by using Monte Carlo simulations. For W  > 1 μm, the MC simulation provides the standard deviation $\sigma_{MC}$ very close to $\sigma_{ddf}$ from the DDF model and to $\sigma_{app}$ from our approximate expression [1].

Previously, we have used the inhouse MC simulation code for Si. However, we do not have the code for Ge. Therefore, the Sentaurus MC simulation code by Synopsys [44] is applied to the analysis on the Ge PD as shown in the next subsection.

The penetration depth δ of 550 nm light in a Ge PD is 20 nm, and 3δ is 60 nm. Therefore, the temporal resolution of Ge PD is analyzed later for the thickness range including these values. The consistency of the analyses in this range by using these two different simulation codes are confirmed for a Si PD. The standard deviations $\sigma_{MC}$ and $\sigma_{MCS}$ of the arrival time for 10 nm < *W* < 100 nm are evaluated and compared, where $\sigma_{MC}$ and $\sigma_{MCS}$ are the standard deviations calculated with our inhouse code and by Sentaurus (with post-processing described in Appendix D). Figure 8 shows the comparison result.

The standard deviations $\sigma_{MC}$ and $\sigma_{MCS}$ agree well around W=100 nm. However, they show a difference of 28% at W=10 nm.

Some other interesting facts are also observed as follows:

the Monte Carlo simulation shows that $\sigma_{MC}$ monotonically decreases when the thickness of the photodiode decreases, while the numerical solution $\sigma_{ddf}$ of the DDF model converges to a constant value for an infinitesimal thickness;for 0.1 μm<W<0.5 μm, $\sigma_{MC}$ departs from $\sigma_{ddf}$, and goes along the approximate expression $\sigma_{app};$
for W<0.1 μm,
$\sigma_{MC}\;\mathrm{resulting}$ from our inhouse MC simulation code departs from $\sigma_{app},$ and, for W<0.03 μm, it becomes parallel to the mixing (drift) component $\sigma_{mixL}=1/\sqrt{12}\;W/v_{c}$ with the slope proportional to W, suggesting that the motion of the signal electrons converges to a ballistic motion (see Appendix C).

### 3.3. Super Temporal Resolution Limit for a Ge PD

We used Sentaurus for the analysis of the Ge PD. A method is developed to estimate the standard deviation of the arrival time from standard outputs from the code (Appendix D).

Figure 9 shows that the theoretical and the practical temporal resolution limits for a Ge PD receiving incident light of 550 nm are, respectively, 0.26 and 1.41 ps for $(W_{T},\sigma$) = (20 nm, 0.13 ps) and $(W_{P},\;3.3\sigma$) = (60 nm, 0.43 ps). The accuracy may be within 30%, considering the result for Si.

The Sentaurus MC simulation of $\sigma_{MCS}$ appears to approach $\sigma_{mixL}\;=1/\sqrt{12}\;W/v_{c},$ representing the mixing effect due to the drift motion and the distribution of the penetration depth. However, in Figure 8 for Si, the standard deviations calculated with our code and the Sentaurus code seem parallel at W=10 nm, showing 28% difference. Further research will make it clear which is correct, asymptotic or parallel approach to $\sigma_{mixL}$. Based on the result, it may be possible to propose an approximate expression of the standard deviation for W<1 μm which is smoothly connected to our approximate expression $\sigma_{app}$ for W>1 μm.

### 3.4. Toward Super Temporal Resolution through SWIR Imaging

At this moment, it is not practical to try to directly achieve the theoretical temporal resolution limit of Ge PDs. The highest $f_{T}$ of transistors is higher than 100 GHz (equivalent to 10 ps). Germanium is a popular material for a photodiode for SWIR imaging up to 1.5 μm light. With some strain to the crystal, the sensitivity can be increased to 1.55 μm at which an optical fiber for communication takes the minimum absorption coefficient. Therefore, we will propose to develop an ultra-high-speed SWIR image sensor of the temporal resolution about 100 ps at first, targeting the STR of 10 ps by gradually thinning the thickness of the Ge PD following the progress of semiconductor technology. It may take a decade as shown in Figure 1. Therefore, the temporal resolution of the Ge PD for the longer wavelength is also analyzed.

Figure 10a,b shows standard deviation results for the Ge PD when incident light with the wavelengths λ of 1 and 1.5 μm. The practical temporal resolutions for the wavelengths are 61.5 ps for 1.58 μm and 180.8 ps for 6.12 μm.

The result suggests development of a SWIR ultra-high-speed image sensor with the thickness of the Ge PD of 1–5 μm at first, which has many important practical applications.

In the SWIR range, the drift motion effect gradually increases. For λ=1 μm in Figure 10a, the diffusion and the drift components are comparable, which makes a slight hump in the shape of the temporal resolution. For λ=1.5 μm in Figure 10b, the drift component is dominant between δ<W<3δ. In these ranges, our approximate expression accurately follows the DDF model, even including the humps. In the range, MC simulations are not applied, since the results may fit our approximate expression as experienced in the simulations for a Si PD.

For the thickness of the photodiode W>1 μm, the temporal resolution limit is accurately calculated by our simple approximate expression. However, for W<1 μm, it is expected for someone to derive a new expression of the standard deviation based on the physics including the ballistic motion, and other important physical components.

## 4. Dark Current

Figure 11 shows a pixel of a backside-illuminated burst image sensors with a Si charge collection pyramid that we previously proposed [45], and with a thin Ge PD. The upper layer of the former structure is a Si PD of about 10 μm thick and the lower layer is a silicon diffusion circuit layer of about 1 μm thick or less. The pyramid efficiently works to collect signal electrons in a very short time with a 100% fill factor. In Figure 11b, the Si PD is replaced by a Ge PD of about 0.1 μm thick. To suppress spread of the arrival time for STR imaging, the lateral size of the backside surface of the Ge PD is about 200 nm, which requires an advanced semiconductor process and an on-chip microlens array.

A Ge PD generates a huge dark current in SWIR imaging at a video rate of 30 fps. However, the dark current during a frame interval of 10 ps for STR imaging is 1/33,333,333,333 of that for video-rate SWIR imaging. Thickness of a Ge PD for STR imaging is about 1/100 of that for SWIR imaging, further reducing the dark current to 1/100. For the same thickness, the dark current is proportional to the square of the lateral size. When the pixel size is about 1 μm, the dark current further reduces to 1/25. Therefore, deep cooling may be unnecessary for the STR image sensor with a Ge PD. Instead, the most important design issue is to efficiently suppress migration of the dark current to the Si circuit layer during transfer of voltage signals converted from signal electron packets in floating diffusions to an analogue or digital memory unit in a functional chip stacked to the sensor chip.

## 5. Concluding Remarks

### 5.1. Conclusions

Super temporal resolution (STR) is defined as a resolution better than the theoretical temporal limit 11.1 ps for silicon. This paper reports a pilot study to achieve STR with an image sensor. Germanium is employed as a candidate photodiode material in the visible light range to show the fundamental feasibility.

Previously, we defined the theoretical temporal resolution limit as $t_{T}=2\sigma \;\mathrm{for}\;W_{T}=\delta$, where σ is the standard deviation of the arrival time of signal electrons, δ is the average penetration depth of incident light, and $W_{T}$ is the thickness of the photodiode [1]. However, the condition results in 16% of temporal crosstalk and 37% loss of incident light passing through the photodiode. Therefore, the practical temporal resolution limit is defined in this paper as $t_{P}=3.3\sigma$ for $W_{P}=3\delta$. For this condition, both the temporal crosstalk and passing-through electrons are both 5%. Then, for silicon, $t_{T}=11.1\;\mathrm{ps}$ and $t_{P}=45.2\;\mathrm{ps}.$

Monte Carlo simulations are applied to an intrinsic Ge PD to estimate these values. The theoretical and the practical temporal resolution limits of the Ge PD are estimated as 0.26 and 1.41 ps, respectively.

For W>1 μm, our simple approximate expression of the standard deviation of the arrival time agrees almost perfectly to the MC simulation results and to the expression based on exact formulation, though, requiring numerical integration. For W<1 μm, a simple approximate expression is expected to be derived.

The dark current is a major issue for SWIR imaging with a Ge PD. However, it is not a serious problem during an image capturing operation with a STR image sensor. Instead, it is a major design issue to efficiently suppress migration of the dark current to the circuit layer after the image capturing operation.

### 5.2. Remarks

The proposed strategy to estimate the temporal resolution limits may be acceptable as a reasonable approximation. However, the values of the temporal resolution limits should be reexamined based on the accurate measurement of the average penetration depths of light which widely vary depending in the literature. We asked Synopsys to improve the Sentaurus MC simulation code for directly calculating standard deviations of the arrival times, which is not supported by the current version. The new version will appear in the near future.

The germanium process has not been fully developed for practical applications. The driver circuit at this moment cannot operate at such a high-speed as the STR image sensor. It is proposed to develop a SWIR ultra-high-speed image sensor with a wider Ge PD of 1 μm thick for NIR light or 5 μm thick for 1.5 μm light as shown in Figure 10, and decrease the size toward the STR along with progress of relating technologies. The SWIR ultra-high-speed image sensor itself has many important practical applications.

It may take a decade to achieve the STR image sensor as predicted in the trend line in Figure 1. Further research with other materials and structures is encouraged.

## Figures and Tables

**Figure 1 sensors-20-06895-f001:**
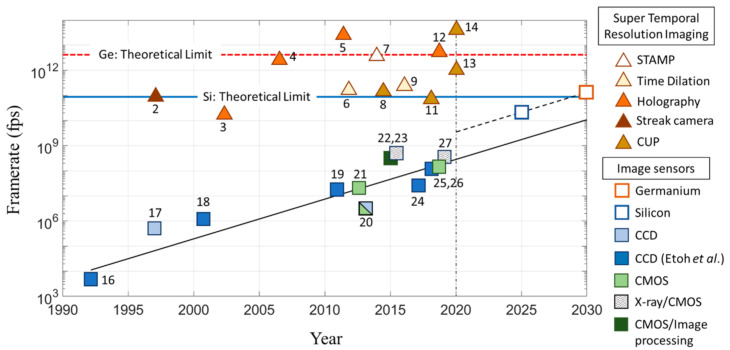
Evolution of ultra-high-speed image sensors and imaging devices having achieved the super temporal resolution. The theoretical temporal resolution limit for a Ge PD is presented later in Section 3.3 of this paper. The slanted dashed line assumes that a chip with in-situ driver circuits is stacked to the sensor chip.

**Figure 2 sensors-20-06895-f002:**
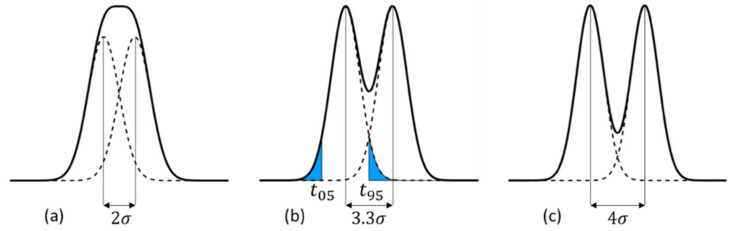
Definitions of the temporal resolution. (**a**) The theoretical temporal resolution limit, $t_{T}=2\sigma$, (**b**) a practical temporal resolution, $t_{P}=t_{95}-t_{05}≅3.3\sigma$, (**c**) for reference.

**Figure 3 sensors-20-06895-f003:**
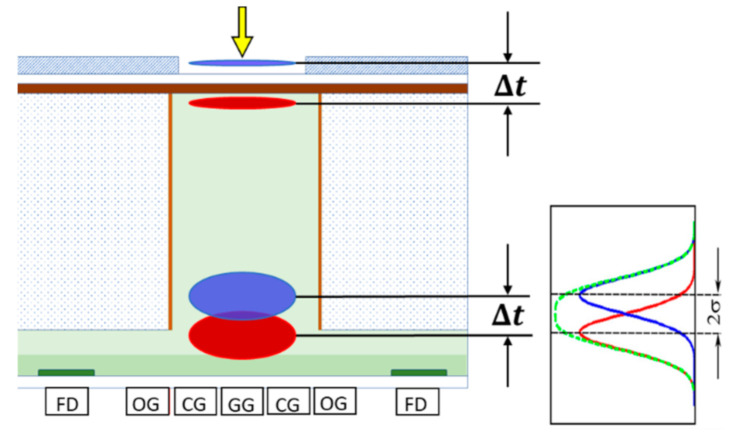
The theoretical temporal resolution limit: $\Delta t=t_{T\;}=2\sigma$.

**Figure 4 sensors-20-06895-f004:**
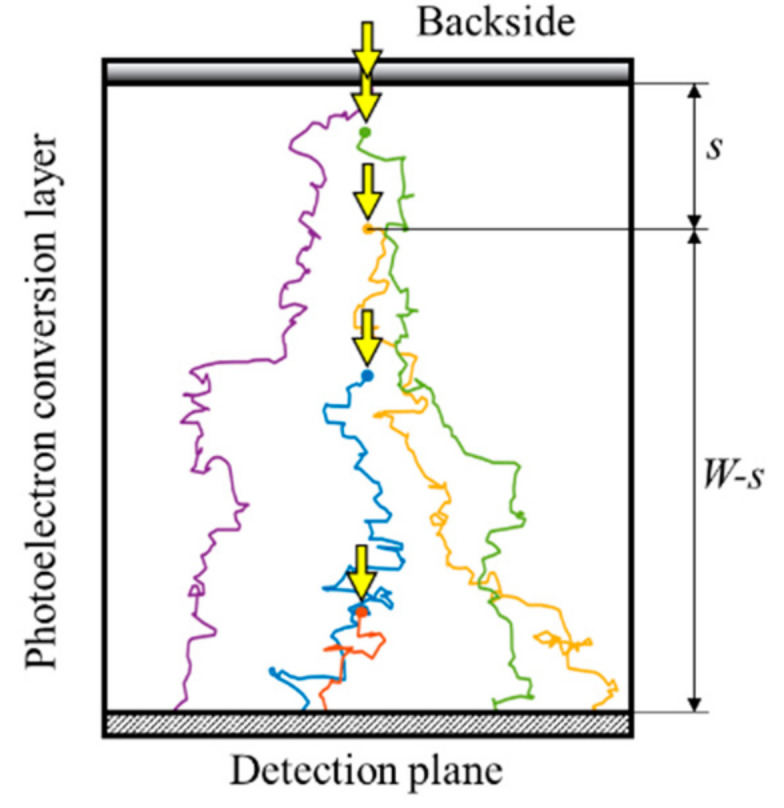
A model to analyze the standard deviation σ of the arrival time of electrons to the detection plane. *W*: thickness of the photodiode, *s*: the generation site of an electron distributing in the depth direction due to the exponential distribution of the penetration depth of light. The bulk is made of an intrinsic silicon. From Figure 5a, the electric fields between the backside and the detection plane are set at 2.5 V/μm for simulations in Si and 0.42 V/μm for simulations in Ge at which the drift velocities are 95% of the fully saturated values (see also Table 1).

**Figure 5 sensors-20-06895-f005:**
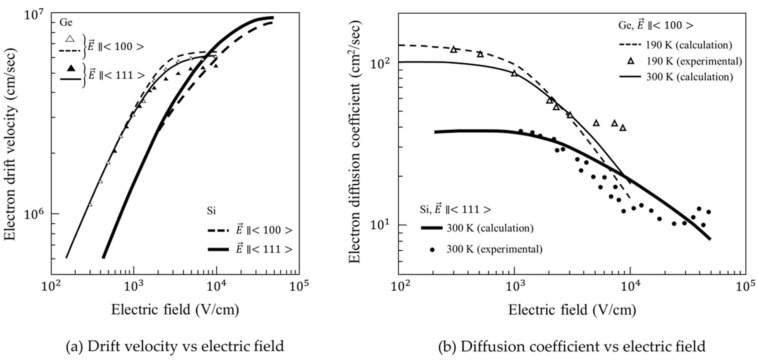
The drift velocity of electrons and the longitudinal diffusion coefficient in Si and Ge: triangles in (**a**) are the experimental data after [36,37], and triangles and circles in (**b**) are the experimental data after [36,38]. Lines are calculated using Monte Carlo simulation codes.

**Figure 6 sensors-20-06895-f006:**
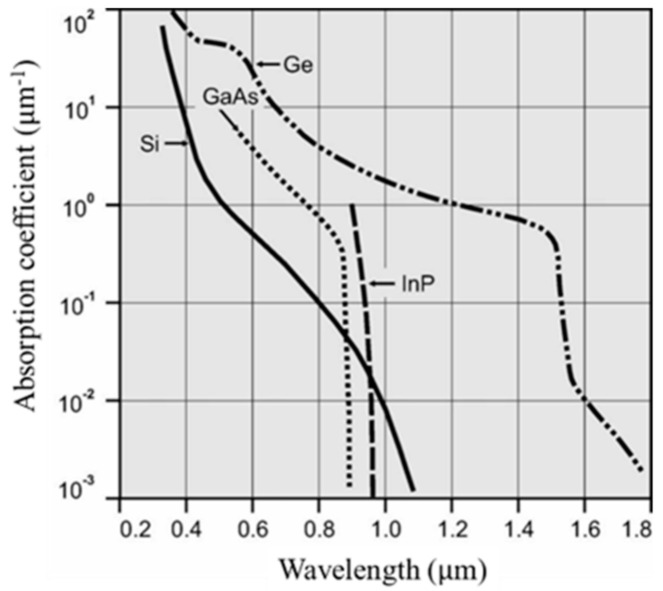
Absorption coefficients for Si, Ge, GaAs and InP [43]. The original date is assembled in [40].

**Figure 7 sensors-20-06895-f007:**
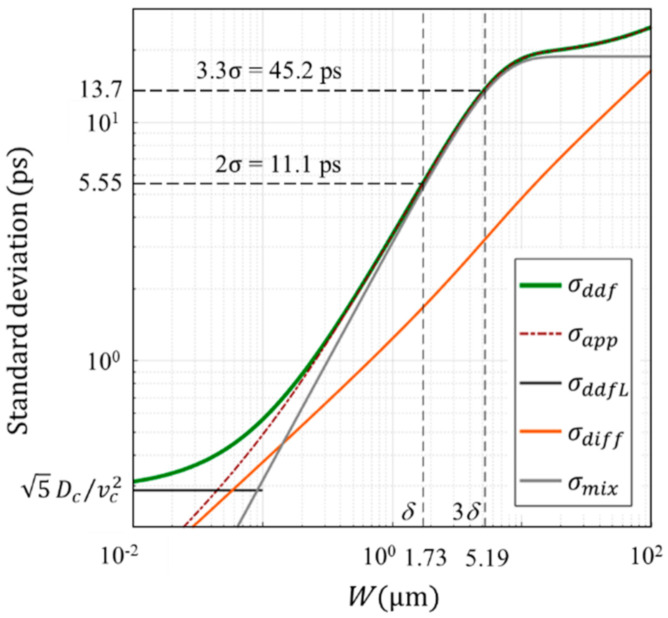
The temporal resolution of the Si PD with incident light of 550 nm along the <111> direction. Solid line $\sigma_{ddf}$: exact formulation of the DDF model; dashed line $\sigma_{app}$: our approximate expression; $\sigma_{ddfL}$: asymptotic value of $\sigma_{ddf}$ for an infinitesimal W; $\sigma_{diff}$ and $\sigma_{mix}$: diffusion and mixing components of
$\sigma_{app}$ where mixing is due to a combined effect of the drift velocity and the distribution of the penetration depth.

**Figure 8 sensors-20-06895-f008:**
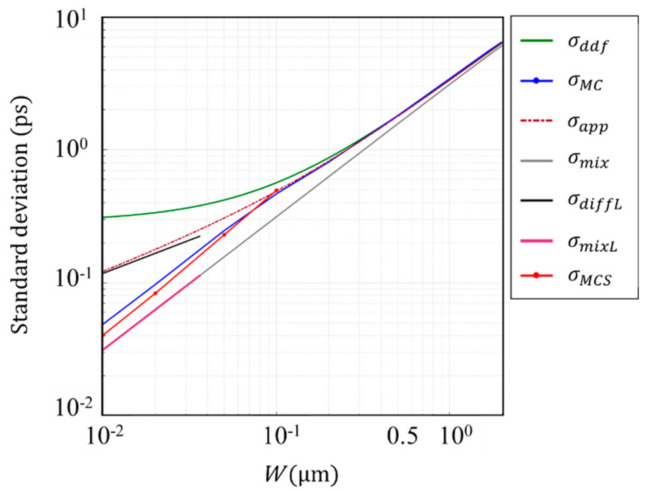
Standard deviation of various relevant parameters for a Si PD for 0.01<W<1 μm. Wavelength of light λ=550 nm; $\sigma_{MC}$ (blue line) and $\sigma_{MCS}$ (red line): standard deviations of the arrival times calculated by our inhouse MC simulation code and by the Sentaurus MC simulation code;
$\sigma_{diffL}$ and $\sigma_{mixL}$; asymptotic expression of $\sigma_{diff}$ and $\sigma_{mix}$; for an infinitesimal *W*; $\sigma_{diff}$ is a straight line proportional to $W^{-1}$, extending $\sigma_{diffL}$ (black line).

**Figure 9 sensors-20-06895-f009:**
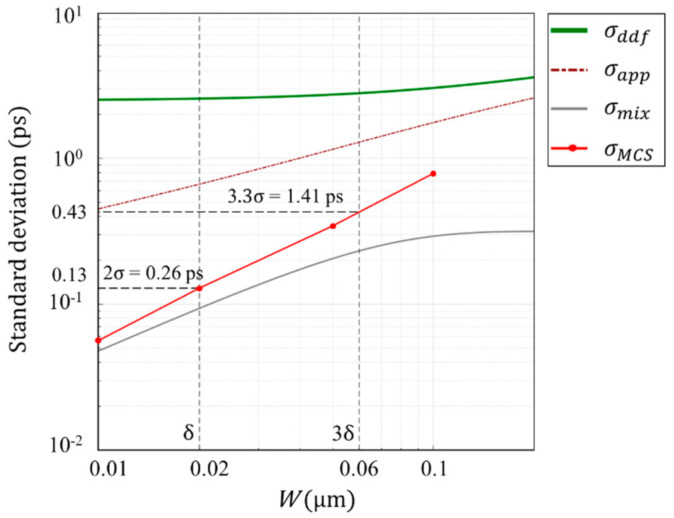
Standard deviation of arrival time and temporal resolutions of the Ge PD for 550 nm illumination with the average penetration depth of 20 nm. $\sigma_{MCS}$ is calculated by a newly proposed method based on Sentaurus MC simulations. The other results in the plots are calculated with equations in Appendix A and Appendix B for the values in Table 1. The diffusion component $\sigma_{diff}$ is not drawn since it completely fits $\sigma_{app}$ for W<0.1 μm for Ge PD.

**Figure 10 sensors-20-06895-f010:**
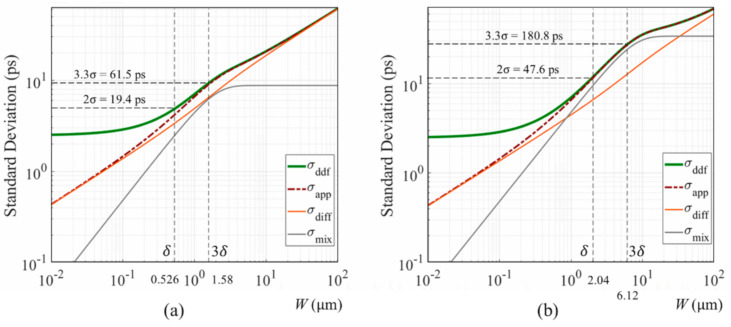
Standard deviations for Ge PDs: (**a**) wavelength of incident light λ=1 μm and the average penetration depth δ=0.526 μm, (**b**) λ=1.5 μm, δ=2.04 μm.

**Figure 11 sensors-20-06895-f011:**
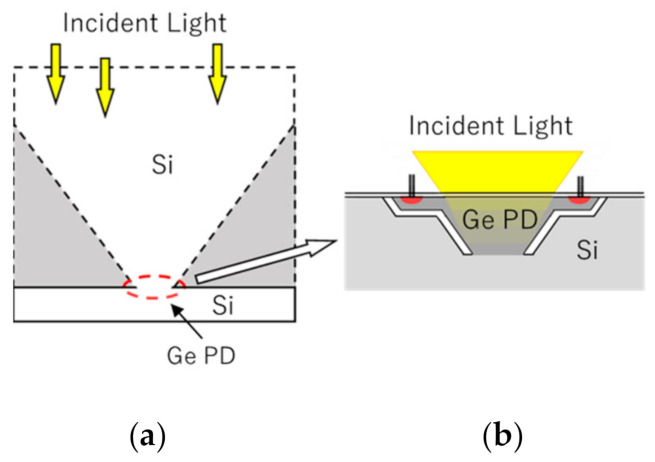
A Si charge collection pyramid (**a**) vs. a Ge PD (**b**).

**Table 1 sensors-20-06895-t001:** Parameters used in the analyses of the standard deviation of the arrival time.

	Si $\vec{E}∥<111>$	Ge $\vec{E}∥<100>$	
Wavelength	0.55 μm	0.55 μm	1 μm
Penetration depth	1.73 μm	20.0 nm	526 nm
Critical E-field	25 kV/cm	4.2 kV/cm	
Drift velocity	9.19 × 10^6^ cm/s	5.8 × 10^6^ cm/s	
Diffusion coefficient	10.8 cm^2^/s	42.5 cm^2^/s	

(1) Si and Ge are intrinsic. (2) The critical field is defined as the field at which the drift velocity is 95% of the saturated value. (3) After figures in [36,38], we measured that, for Si, the diffusion coefficients at the critical fields are 13.1, 12.1, 10.8, and 10.8 cm^2^/s for 77, 160, 200, and 300 K, respectively. They take close values for 200 and 300 K. For Ge, the diffusion coefficients at critical fields are 48.5, and 42.5 cm^2^/s at 77 and 190 K. Therefore, the diffusion coefficient for Ge at 300 K is assumed to be 42.5 cm^2^/s.

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
