# Peer review of "Toward the Super Temporal Resolution Image Sensor with a Germanium Photodiode for Visible Light"

_sensors, 2020, doi:10.3390/s20236895_

Round 1
Reviewer 1 Report
The topic of high-speed imaging is certainly important and hot, and exploratory ideas for future improvements are welcome. From the paper title: "Toward the Super Temporal Resolution Image Sensor with a Germanium Photodiode for Visible Light" I would expect to read a "roadmap" paper, also addressing the design and technological challenges towards the final objective.
But, as the authors write (line 42) : "This paper is a preliminary report toward achieving STR." In my opinion, this is indeed too preliminary to deserve publication at this stage, because of its insufficient significance and contribution.
The paper has mainly the flavour of a mathematical exercise (not so clearly explained and easy to follow, also due to the use of 4 appendixes), and seems to lack a solid tie to most practical problems for an implementation.
As an example, thinking about an image sensor, I wonder how the pixel read-out electronics should be organised to preserve the photodiode temporal resolution below 1ps; Will germanium require cooling rather than being operated at room temperature in order to limit the dark current ? What about the technological approach ? Etc. But these and other aspects of practical importance are not touched in the paper.
The paper remains only on the theoretical side (and this should be reflected in a different title), trying to provide as an ultimate result the estimate of the temporal resolution. But the authors write (lines 327-330): "The proposed strategy to estimate the temporal resolution limits may be acceptable as a reasonable approximation. However, the values of the temporal resolution limits should be reexamined based on the accurate measurement of the average penetration depths of light which widely vary depending in the literature."
The uncertainty on the results makes the importance of the paper very limited.
Since the case of silicon has already been theoretically treated by the authors in reference [1] (that, by the way, is written much more clearly), here I would expect the case of Germanium to be covered more thoroughly.
But the Monte Carlo approach adopted is not described in a clear way and is not validated. A cross-verification between two codes is only applied to the case of silicon, showing fairly good agreement, but this does not support the quality of the simulation in case of Germanium. Isn't it possible to validate the simulation against some experimental results (albeit referred to a thicker Germanium sensor)?
In another part (lines 274-275), referred to the Ge photodiode, it is written:
"A further analysis is necessary to make an expression of the standard deviation of the arrival time for ?<1 μm."
But from the paper title and the abstract one would expect the case of a very thin Ge photodiode for visible light to represent the main goal of the authors, in perspective. This is clearly another weak point.
Author Response
Thank you very much for your review of our paper. Please see the attachment for my answers.

Reviewer 2 Report
This paper is an initial theoretical and simulation investigation in the possible fundamental limits in signal timing for semiconductor detectors, specifically for very high frame-rate imaging applications. The paper is logically organized and well written. The authors clearly discuss the criteria for evaluation of performance, introducing practical requirements, more useful for interpretation than purely theoretical ones. The methods are described in details and are first validated for Si sensors, with previously established results. Then, the results for Ge sensors of relevant thicknesses are presented, showing exciting potential.
The article is of high quality, but could still benefit from a few minor improvements:
- In introduction and conclusion, it would be very useful to discuss briefly more on applications, and importance of improving temporal resolution, especially to such high temporal resolution (maybe after line 76, where 1/30 improvement is estimated). For example, what would such a fast (~1ps) imaging sensor, with such high FPS (~Tfps), be used for, and what would it enable that is currently not possible.
- Line 26: “of Ge for the wavelength”: which wavelength?
- 171-182: can there be some comment, why is absorption length so poorly known?
- 186: what is meant by professional society?
- 223-228: This text repeats too many times through the article, for example again at 151-157 and 255-259. Also, a brief description of the approach explained in Appendix D would be welcome already at some point in the text (suggest at line 157).
- 298 and 323: Last sentence (“is expected to derive...”) unclear: will you derive the expression?/is it expected the the expression will be simple once derived?/...
Some language suggestions:
24: along the progress -> along with the progress
123: agrees the -> agrees with the
298: “However, W > 1mum,” -> “However, for W > 1mum,”
Author Response

(The authors gave the same response as above.)

Round 2
Reviewer 1 Report
Dear Authors,
Thank you for addressing my comments and requests.
The paper and its scope are now more clear. No further comment from my side.
Best wishes
Your reviewer